# Personalizing Clozapine in Treatment-Resistant Schizophrenia: The Role of MicroRNA Biomarkers—A Pilot Study

**DOI:** 10.3390/cimb47121020

**Published:** 2025-12-07

**Authors:** Dmitry N. Sosin, Aiperi K. Khasanova, Roman A. Illarionov, Anastasia K. Popova, Karin B. Mirzaev, Andrey S. Glotov, Sergey N. Mosolov, Dmitry A. Sychev

**Affiliations:** 1Psychiatric Hospital No. 1 Named After N.A. Alexeev of the Department of Health of Moscow, 117152 Moscow, Russia; 2Federal State Budgetary Research Institution Russian Research Center of Surgery Named After Academician B.V. Petrovsky, 119991 Moscow, Russia; karin05doc@yandex.ru (K.B.M.); dmitry.alex.sychev@gmail.com (D.A.S.); 3Federal State Budgetary Educational Institution of Further Professional Education “Russian Medical Academy of Continuous Professional Education”, Ministry of Healthcare, Russian Federation, 125993 Moscow, Russia; abdyrahmanova_peri@mail.ru (A.K.K.); profmosolov@gmail.com (S.N.M.); 4D. O. Ott Research Institute of Obstetrics, Gynecology and Reproductology, 199034 Saint-Petersburg, Russia; raillarionov@gmail.com (R.A.I.); stassi1997@mail.ru (A.K.P.); anglotov@mail.ru (A.S.G.); 5Department of Genetics and Biotechnology, Saint Petersburg State University, 199034 Saint-Petersburg, Russia; 6Moscow Research Institute of Psychiatry, Branch of the V.P. Serbsky National Medical Research Center for Psychiatry and Narcology, Ministry of Health, Russian Federation, 107076 Moscow, Russia

**Keywords:** treatment-resistant schizophrenia, psychiatry, clozapine, miRNA, epigenetics

## Abstract

Background: Clozapine remains the only antipsychotic with proven efficacy in treatment-resistant schizophrenia (TRS). However, it is effective in only about 40% of patients and is associated with numerous adverse drug reactions. Personalization of clozapine therapy is therefore of critical importance in clinical psychiatry. MiRNA expression may serve as a promising exploratory marker for understanding individual variability in clozapine efficacy and safety. Methods: In this study, we determined the complete miRNA expression profile in TRS patients before initiation of clozapine and after four weeks of treatment. Results: In 15 inpatients with TRS receiving 4-week clozapine monotherapy, PANSS total decreased from 98.8 ± 13.19 to 80.47 ± 14.63 (*p* = 0.001). The most frequent adverse drug reactions were hypersalivation (*n* = 13), drowsiness/sedation (*n* = 12), and prolonged sleep (*n* = 12). We detected 24 differentially expressed miRNAs after clozapine. Changes in hsa-miR-129-5p, hsa-miR-6068, and hsa-miR-6814-5p correlated with improvements in positive symptoms; hsa-miR-128-1-5p tracked general psychopathology; and hsa-miR-6814-5p aligned with global improvement (lower PANSS total, higher PSP). Safety signals included associations of hsa-miR-4472 with asthenia/fatigue and prolonged sleep, hsa-miR-4510 with prolonged sleep, hsa-miR-615-3p and hsa-miR-4715-3p with tachycardia, and hsa-miR-329-1-5p with weight gain. Conclusions: Because miRNAs regulate the expression of a wide range of genes, including those involved in clozapine’s efficacy and safety, these findings underscore the need for further studies integrating pharmacoepigenetic and pharmacogenetic biomarkers. Our preliminary findings suggest that specific miRNAs could be candidate biomarkers associated with clozapine response in TRS, although these results require validation in larger and controlled studies.

## 1. Introduction

Schizophrenia is a severe psychiatric disorder that markedly impairs social and occupational functioning [1]. Despite major advances in psychopharmacology, effective treatment remains a central challenge in clinical practice [2,3]. About one-third of patients develop treatment resistance over the course of illness [4]. Treatment-resistant schizophrenia (TRS) places a heavy burden on healthcare systems and society [5]; affected patients experience greater disability and incur substantially higher treatment costs than responders [6].

Clozapine is the only antipsychotic with proven efficacy in TRS [3]; however, only about 40% of patients achieve a clinical response. Delayed initiation in TRS is associated with a markedly higher risk of treatment failure [7], whereas early initiation substantially improves outcomes, underscoring the need to predict both clozapine efficacy and safety [8].

In recent years, pharmacoepigenetics—particularly the study of miRNAs (miRNAs) as biomarkers for psychiatric disorders—has attracted growing interest [9,10]. miRNAs are small non-coding RNAs that regulate transcript stability and translation, thereby shaping broader gene-regulatory networks [11]. They act primarily by binding to the 3′ untranslated regions (UTRs) of target messenger RNAs [12]. A single miRNA can regulate numerous genes, and bioinformatic tools help identify putative targets; for example, according to miRDB (https://mirdb.org/mirdb/index.html, accessed on 20 November 2025), hsa-miR-3183 is predicted to regulate 498 genes. Some of these genes relate to clozapine’s mechanism of action, such as GRINA. Experimental evidence links clozapine efficacy to enhanced NMDA receptor activity, a finding supported by animal studies [13,14,15].

Defining miRNA profiles linked to clozapine efficacy and safety is a key step toward personalizing treatment for TRS. Given the exploratory scope of this work, we performed a pilot study to identify candidate miRNAs in a small TRS cohort. This approach supports hypothesis generation and lays the groundwork for future large-scale validation.

## 2. Materials and Methods

Inclusion criteria were: (1) age between 18 and 65 years; (2) a diagnosis of schizophrenia (F20) according to ICD-10; (3) disease duration of at least 24 months; (4) current exacerbation of psychosis; and (5) lack of response to at least two adequate antipsychotic trials (≥4 weeks each, at least one involving an atypical antipsychotic at a minimum dose equivalent to 400 mg of chlorpromazine). The study workflow is shown in Figure 1.

Exclusion criteria included (1) comorbid psychiatric disorders (including current alcohol or substance dependence), (2) decompensated somatic or infectious diseases, or a history of brain injury associated with loss of consciousness or seizures. All patients were treated as psychiatric inpatients in 2022.

The ICD-10 diagnosis of schizophrenia (F20) was verified using the MINI 5.0 structured diagnostic interview [16]. At baseline, psychopathological symptom severity was assessed with the PANSS and CGI-S scales [17,18], and functioning was evaluated using the PSP scale [19].

The study included 15 male inpatients with schizophrenia who met the above criteria. All patients received clozapine monotherapy. Doses were individually titrated in accordance with clinical guidelines, with the objective of achieving the minimum effective dose. Daily clozapine doses ranged from 150 mg to 550 mg, with a mean final dose of 340 ± 123.49 mg/day.

All participants were continuously hospitalized in the same unit throughout the study and received hospital-provided meals and standardized daily schedules. Clozapine was administered as monotherapy. Blood collection, processing, and storage conditions were uniform across patients and visits.

A follow-up visit was conducted after 28 ± 3 days to reassess the PANSS, CGI-S, CGI-I, and PSP scales, and to evaluate treatment safety using the UKU scale [20]. All 15 patients completed the study. The mean patient age was 36 ± 6 years, with a median age at schizophrenia onset of 17 ± 5 years. The mean duration of illness prior to study entry was 18 ± 9 years.

Venous blood samples were collected from each patient at baseline into K2EDTA Intravacuate tubes and repeated at the second visit. Samples were centrifuged at 2500 rpm for 10 min at 4 °C. Plasma was then aliquoted into cryotubes, frozen, and stored at −80 °C until analysis no more than 4 months.

miRNA was isolated from 200 μL of plasma using the miRNeasy Serum/Plasma Kit (Qiagen, Hilden, Germany) according to the manufacturer’s instructions. The miRCURY RNA Spike-In Kit (Qiagen) were used for quality control of the RNA extraction and further amplification. The NanoDrop 2000 micro-volume spectrophotometer (Thermo Fisher Scientific, New York, NY, USA) was used to estimate the concentration and purity of the obtained RNA. Library yield, fragment distribution, and molar concentration were assessed using the Agilent 2200 TapeStation system with High Sensitivity D1K ScreenTape reagents (Agilent Technologies, Santa Clara, CA, USA). Sequencing quality control parameters (read depth, mapping rates, and duplication rates) were verified by the sequencing facility as part of the Qiagen GeneGlobe workflow and confirmed to be within the standard acceptable range for small RNA-seq runs. RNA was resuspended in 12 μL of RNase-free water and stored at −80 °C until library preparation.

Libraries were generated with the QIAseq miRNA Library Kit (Qiagen, Germany). RNA samples were ligated to 3′ and 5′ adapters, followed by reverse transcription. cDNA was purified using magnetic beads and eluted in 17 μL of nuclease-free water. PCR amplification was performed with unique barcodes to enable pooling of libraries before sequencing. Amplified products were purified with magnetic beads and eluted in 25 μL of nuclease-free water. Library yield, fragment distribution, and molar concentration were assessed using a 2200 TapeStation system with High Sensitivity D1K ScreenTape and reagents (Agilent Technologies, Santa Clara, CA, USA). The number of libraries required for sequencing was determined according to the manufacturer’s recommendations. Sequencing was carried out on a HiSeq 2500 platform (Illumina, San Diego, CA, USA) with 75 bp single-end reads.

Small-RNA sequencing data were processed using the GeneGlobe Data Analysis Center (Qiagen). Differential expression analysis of raw counts (30 samples) was conducted in DESeq2 with median-of-ratios normalization [21]. miRNAs were considered differentially expressed if the adjusted *p*-value was < 0.05 and |log2FC| > 1.5. Pathway enrichment analysis of significant miRNAs was performed using DIANA-miRPath v4.0 [22].

miRNA nomenclature was standardized according to the official miRBase database (Release 22.1; https://www.mirbase.org/, accessed on 20 November 2025) [23], ensuring consistency across the manuscript. All miRNAs were standardized according to miRBase Release 22.1. Locus-specific variants were corrected to the official mature forms (e.g., hsa-miR-329-1-5p was corrected to hsa-miR-329-5p). For sequences not currently included in miRBase (e.g., hsa-miR-7853-3p), the identifiers provided by the sequencing pipeline were retained.

Statistical analyses were performed with SPSS Statistics 26.0 (IBM, Armonk, NY, USA). Quantitat ive variables are presented as mean ± standard error, and categorical variables as counts. Parametric tests included Student’s *t*-test (two groups) and one-way ANOVA (≥3 groups). Non-parametric tests included the Mann–Whitney U test (two groups) and the Kruskal–Wallis test (≥3 groups). Between-group differences (e.g., ΔmiRNA in patients with vs. without a given ADR) were tested with the Mann–Whitney U test. Given the deviation from normality (Shapiro–Wilk test) and the limited subgroup sizes, comparisons were primarily conducted using the Mann–Whitney U test.

Categorical distributions were compared using Pearson’s chi-square test, with Fisher’s exact test applied for 2 × 2 comparisons. Multiple comparisons were adjusted with the Bonferroni correction. For miRNA screening, Bonferroni-adjusted significance thresholds (*p*_adj_) were applied. As this was a pilot, hypothesis-generating study, subsequent analyses were performed using uncorrected *p*-values. Findings should therefore be regarded as preliminary and interpreted with caution pending external validation. A *p*-value < 0.05 was considered statistically significant.

## 3. Results

### 3.1. Efficacy of Clozapine Therapy

The mean score on the PANSS positive symptoms subscale decreased from 23.67 ± 4.05 (visit 1) to 16.2 ± 4.89 (visit 2) (*p* = 0.001). The score of negative symptoms decreased from 29.2 ± 5.92 (visit 1) to 27.27 ± 6.08 (visit 2) (*p* = 0.006).The score of the general psychopathological symptoms subscale decreased from 45.93 ± 7.19 to 37.67 ± 7.58 (*p* = 0.002). Overall, there was a reduction in the total PANSS score from 98.8 ± 13.19 to 80.47 ± 14.63 (*p* = 0.001).

The CGI-S scale score decreased from 5.8 ± 0.41 to 4.6 ± 0.83 (*p* = 0.0001). According to the CGI-I scale, two patients showed no improvement in psychotic symptoms, 7 patients showed minimal improvement, and 6 patients showed significant improvement.

There was also an increase in the mean PSP scale score from 34.07 ± 9.13 to 49.73 ± 13.03 (*p* = 0.002), indicating an improvement in social and daily functioning.

### 3.2. Safety of Clozapine Therapy

The safety of clozapine therapy was assessed using the UKU scale. The following adverse drug reactions (ADRs) were observed during clozapine therapy. The most common side effects included hypersalivation (*n* = 13); drowsiness and/or sedation (*n* = 12); increased sleep duration (*n* = 12); asthenia, somnolence and fatigue (*n* = 9); orthostatic hypotension (*n* = 9); and constipation (*n* = 5). Relatively rare side effects included heart palpitations, tachycardia (*n* = 3); weight gain (*n* = 3); eye accommodation disorder (*n* = 2); dry mouth (*n* = 2); weight loss (*n* = 2); decreased concentration (*n* = 1); emotional indifference (*n* = 1); dystonia (*n* = 1); tremor (*n* = 1); akathisia (*n* = 1); and urinary difficulties (*n* = 1).

### 3.3. Characteristics of MiRNAs

A total of 1917 miRNAs were detected. The expression of 24 miRNAs changed following clozapine administration. Table 1 summarizes the differentially expressed miRNAs detected by DESeq2 (median-of-ratios), using *p*_adj_ < 0.05 and |log2FC| > 1.5. We applied stringent selection criteria for significantly expressed miRNAs: *p*_adj_ < 0.05; log2FoldChange ≥ 1.

### 3.4. Relationships Between Clinical Parameters and miRNA Expression

For all subscales of the PANSS, as well as the CGI-S, PSP scales, and miRNA expression levels, delta changes in scores between visits were calculated (visit 2 minus visit 1). The associations between these deltas were then evaluated.

Relationships between quantitative variables were assessed using correlation analysis, yielding the following results.

The delta scores of the PANSS positive symptoms subscale were negatively correlated with delta changes in the expression levels of hsa-miR-129-5p (r = −0.511; *p* = 0.05), hsa-miR-6068 (r = −0.564; *p* = 0.029), and hsa-miR-6814-5p (r = −0.511; *p* = 0.05).

The delta scores of the PANSS general psychopathology subscale were positively correlated with delta changes in the expression levels of hsa-miR-128-1-5p (r = 0.585; *p* = 0.022).

The delta total PANSS score was negatively correlated with delta changes in the expression levels of hsa-miR-6814-5p(r = −0.556; *p* = 0.031).

The delta scores of the PSP scale were positively correlated with delta changes in the expression levels of hsa-miR-6814-5p (r = 0.658; *p* = 0.008).

The relationship between ADRs and changes in miRNA expression levels was examined using the Mann–Whitney U test. Only the presence of ADRs was considered, not their severity.

Δhsa-miR-4472 was higher in patients with asthenia, somnolence, and increased fatigue (*n* = 9 vs. 6; *p* = 0.036) than in those without (Figure 2).

Δhsa-miR-4472 (*p* = 0.048) and Δhsa-miR-4510 (*p* = 0.048) was higher in patients with increased sleep duration (*n* = 12 vs. 3) than in those without (Figure 3 and Figure 4).

Δhsa-miR-615-3p (*p* = 0.048) and Δhsa-miR-4715-3p (*p* = 0.048) was higher in patients with tachycardia (*n* = 3 vs. 12) than in those without (Figure 5 and Figure 6).

Δhsa-miR-329-5p (*p* = 0.048) was lower in patients with weight gain (*n* = 12 vs. 3) than in those without (Figure 7).

Taken together, 4-week clozapine monotherapy produced clinically meaningful improvements in illness severity and functioning, alongside a characteristic ADR profile. Within this context, a small panel of circulating miRNAs showed coherent relationships with both efficacy (e.g., hsa-miR-129-5p, hsa-miR-6068, hsa-miR-6814-5p, hsa-miR-128-1-5p) and safety endpoints (e.g., hsa-miR-4472, hsa-miR-4510, hsa-miR-615-3p, hsa-miR-4715-3p, hsa-miR-329-1-5p), suggesting that dynamic miRNA changes may index treatment response and adverse effects in TRS. These signals warrant validation in larger, controlled cohorts.

## 4. Discussion

We report preliminary findings from a prospective study evaluating the full miRNA expression profile as a potential predictor of the efficacy and safety of 4-week clozapine monotherapy in 15 patients with TRS. These findings are preliminary, and larger, rigorously designed studies are needed.

Twenty-two miRNAs showed altered expression after clozapine administration. We found correlations with clozapine efficacy for hsa-miR-129-5p, hsa-miR-6068, hsa-miR-6814-5p, and hsa-miR-128-1-5p. We observed associations with clozapine safety for hsa-miR-4472, hsa-miR-4510, hsa-miR-615-3p, hsa-miR-4715-3p, and hsa-miR-329-5p. Table 2 summarizes the significant associations between changes in miRNA expression and the clinical endpoints of efficacy and safety.

Our study confirmed clozapine’s efficacy in TRS, driven mainly by reductions in positive and general psychopathology on the PANSS. We also noted a non-significant decrease in negative symptoms. Improvements on the CGI-S and CGI-I further supported clozapine’s effect. In addition, overall CGI improvement coincided with gains in professional and social functioning on the PSP.

We found that rising hsa-miR-6814-5p levels during clozapine treatment signaled treatment efficacy. Patients with higher plasma hsa-miR-129-5p, hsa-miR-6068, and hsa-miR-6814-5p showed greater improvement in positive symptoms. When hsa-miR-128-1-5p increased modestly, clozapine more effectively reduced general psychopathology.

The ADR profile in our cohort was consistent with prior reports [24]. In analyses of ADR–miRNA relationships, higher hsa-miR-4472 expression was associated with asthenia; longer sleep duration was associated with higher hsa-miR-4472 and hsa-miR-4510 expression; tachycardia was associated with higher hsa-miR-615-3p and hsa-miR-4715-3p expression; and weight gain was associated with lower hsa-miR-329-5p expression

Recent work underscores the promise of miRNAs as biomarkers for schizophrenia and for antipsychotic efficacy [25,26]. Pérez-Rodríguez et al. (2023) identified 16 miRNAs that distinguished TRS patients from responders [9]. Another study reported differential expression of hsa-miR-181b-5p, hsa-miR-195-5p, and hsa-miR-301a-3p between TRS and responder groups [27]. A further investigation with a similar design found differences in 34 miRNAs between these cohorts [28]. A recent systematic review showed consistent overexpression of miR-181b-5p and miR-34a-5p in individuals with schizophrenia compared with healthy controls [29]. The miRNAs we identified differ from those reported previously, likely reflecting differences in study design and in the specific comparisons made (TRS vs. responders and/or healthy controls).

In a separate study, clozapine increased hsa-miR-675-3p expression in patients with TRS [30]. The authors used a cross-sectional comparative design and additionally tested the effect in vitro in neuroblastoma cells. By contrast, in our study hsa-miR-675-3p expression did not change during clozapine treatment. These discrepancies likely reflect differences in study design between our work and that of Funahashi et al. (2023) [30].

Preclinical data indicate that antipsychotics influence epigenetic regulation—including long non-coding RNAs, DNA methylation, and miRNAs [31]. This line of research holds strong promise for clarifying schizophrenia pathogenesis and the mechanisms of antipsychotic action. Major challenges remain, as the epigenetic landscape reflects many influences, including disease pathophysiology, treatment duration, biochemical status, genetic predisposition, and environmental exposures. These considerations underscore the need for further large-scale clinical studies.

Other studies have shown that hsa-miR-143-3p and hsa-miR-195 have predictive value for treatment efficacy in patients with schizophrenia (non-TRS) [32]. Associations with schizophrenia have also been reported for hsa-miR-570-3p, hsa-miR-550a-3p, hsa-miR-130a-3p, hsa-miR-210, hsa-miR-337-3p, and hsa-miR-130b-3p, although TRS was not analyzed as a distinct subgroup in those studies [33].

In addition, several miRNAs whose expression changed with clozapine in our study have been linked to neuropsychiatric disorders. Specifically, hsa-miR-128a/b, hsa-miR-4510, hsa-miR-615-3p, and hsa-miR-4449 have been associated with Alzheimer’s disease or frontotemporal dementia [34,35,36,37,38,39,40,41].

Lower plasma levels of hsa-miR-363 have been reported in patients with anxiety and depression comorbid with substance use disorders [42]. Elevated hsa-miR-4449 levels have also been identified in fibroblasts from individuals with schizophrenia [43].

The interplay between genetic and epigenetic factors is critical to understanding schizophrenia risk and predicting treatment outcomes [44]. Diverse mechanisms shape the functional expression of genetic information [45]. Recent work shows that multi-omics algorithms can improve prediction of antipsychotic drug safety [46]. Against this backdrop, identifying miRNAs linked to clozapine efficacy and safety is a timely and clinically relevant goal.

We conducted a KEGG pathway analysis of miRNA-mediated mechanisms using DIANA-miRPath v4.0 with the TargetScan v8.0 database [22]. The databases were accessed on 10 August 2025. Table 3 reports KEGG pathway enrichment for the identified miRNAs (key pathways and *p*-values). In our study, hsa-miR-129-5p, hsa-miR-6068, hsa-miR-6814-5p, and hsa-miR-128-1-5p were associated with treatment efficacy, and we evaluated their pathways collectively. These four miRNAs converged on two pathways: axon guidance and pathways in cancer. Because the latter is not directly relevant to psychiatric mechanisms, we focused on axon guidance. As shown by Wang et al. (2018), this pathway relates to schizophrenia through its roles in neural progenitor proliferation and differentiation, as well as neuronal migration and positioning [47].

Next, we examined the miRNAs associated with safety parameters, evaluating each miRNA individually. As summarized in Table 3, most enriched pathways relate to the regulation of cell growth, proliferation, differentiation, and survival across multiple cell types, including axon guidance, ErbB signaling, Ras signaling, pathways regulating stem-cell pluripotency, Wnt signaling, neurotrophin signaling, and FoxO signaling. All of these pathways have been implicated in schizophrenia [48,49,50,51,52,53].

These miRNAs also converged on the calcium signaling pathway, which has been linked to schizophrenia [54]. Clozapine itself significantly alters calcium homeostasis in the rat cerebral cortex [55], pointing to a potential role for calcium signaling in both its efficacy and safety.

Clozapine is known to act, in part, as an α2-adrenergic receptor antagonist [56]. Accordingly, the link between the adrenergic signaling in cardiomyocytes pathway and its safety profile (e.g., increased sleep duration) is consistent with clozapine’s established pharmacology.

Finally, the apelin signaling pathway has been implicated in schizophrenia, supported by reports of altered apelin levels in patients versus healthy controls [57,58,59].

Interestingly, hsa-miR-329-1-5p, which in our study was linked to metabolic disturbances, also participates in metabolic pathways—including prolactin signaling and cholesterol metabolism—whose dysregulation has been implicated in obesity [60].

These findings indicate that miRNAs exert broad regulatory effects across multiple physiological systems. The underlying processes are highly complex and reflect both genetic and epigenetic mechanisms. Future work should prioritize identifying and validating the relevant target genes. Combining pharmacogenetic and pharmacoepigenetic markers may, pending validation in larger cohorts, support the development of tools to personalize clozapine therapy. Larger, methodologically rigorous studies with appropriate control groups are needed to clarify the role of the identified miRNAs in clozapine efficacy and safety, ultimately enabling algorithms that guide personalized treatment selection.

For miRNAs linked to treatment efficacy, hsa-miR-6068 and hsa-miR-129-5p are predicted to regulate genes spanning synaptic and neuronal homeostasis, including *CHST3* (extracellular matrix sulfation), *LYNX1* (cholinergic/nicotinic receptor modulation), *KCTD13* (neurodevelopmental signaling), *SLC9A8* (endosomal/vesicular pH control), *IQCE* (cilia-related signaling), and *ATG16L2* (autophagy) [22]. This constellation converges on processes that shape synaptic transmission and plasticity—from cholinergic tone and receptor gating to extracellular matrix remodeling and intracellular trafficking—which are mechanistically compatible with clozapine’s effects on cortical circuitry [13,14,15]. Taken together, these gene-level links provide a plausible substrate for the observed clinical improvement and complement our pathway-level enrichment (e.g., axon guidance) [47], while remaining preliminary and in need of validation in larger cohorts.

For efficacy-linked miRNAs, hsa-miR-6814-5p and hsa-miR-128-1-5p jointly map to targets involved in neuronal metabolism, proteostasis, and neurotransmission, including *RDH16* (retinoid metabolism), *MARS* (translation/methionine charging), *TSPAN11* (membrane microdomains), *SH2D7* (signal transduction), *CDRT1*/*TRIM16*/*CMT1A* region transcript (ubiquitin–proteasome regulation), *FBXO7* (E3-ligase complex/mitophagy), and *SLC6A20* (proline/iminoglycine transporter influencing glutamatergic tone) [22]. This constellation implicates retinoid signaling, protein quality control, and amino-acid transporter–dependent synaptic homeostasis—mechanisms compatible with clozapine’s network-level effects and our axon-guidance enrichment [13,14,15,47]. The overlap at proteostasis and membrane organization suggests convergent regulation that could contribute to clinical improvement. These gene-level links remain preliminary and require validation in larger cohorts.

For the adverse event of prolonged sleep duration, the miRNAs hsa-miR-4472 and hsa-miR-4510 converge on genes governing synaptic release, calcium homeostasis, and vesicular trafficking: *UNC13A* (vesicle priming), *SYT5* (Ca^2+^-dependent exocytosis), *STIM1* (ER Ca^2+^ sensing), *TP6V0E2* (vesicle acidification), *KLC4* (axonal transport), and *SCAMP2* (secretory membrane complexes) [22]. Modulation of *DVL3*/*RSPO4* (Wnt signaling) and *PIK3R2*/*AGAP2* (PI3K/ARF pathways) points to adjustments in neuronal excitability and network plasticity that are consistent with a sedative/somnolent profile [24]. Additional nodes: *TTBK1* (tau kinase), *KMT2D* (epigenetic methyltransferase), *TULP3* (ciliogenesis/trafficking), *SLC25A44* (mitochondrial carrier), and *GDF11* (growth/differentiation)—implicate axonal/ciliary signaling, cellular energetics, and epigenetic control in sleep–wake regulation [54,55]. Taken together, this gene-level architecture supports the biological plausibility of miRNA-linked sleep prolongation during clozapine treatment, while remaining preliminary and in need of replication with quantitative sleep measures [24].

The miRNAs associated with clozapine-related tachycardia, hsa-miR-615-3p and hsa-miR-4715-3p, are predicted to target *SEMA4B* (semaphorin signaling; axon-guidance/autonomic regulation) and *ZNF626* (a zinc-finger transcriptional regulator) [22]. This pattern suggests a contribution of semaphorin-mediated neuronal signaling and transcriptional programs to autonomic heart-rate control; these observations are preliminary and warrant validation with cardiac covariates and clozapine exposure accounted for [24].

The link between hsa-miR-329-5p and clozapine-associated weight gain is mechanistically plausible: predicted targets span *GPC4* (adipokine/insulin-related signaling), *GATM* (creatine metabolism), *MRPL19* and *SLC25A38* (mitochondrial bioenergetics/heme synthesis), together with ubiquitin–proteasome/inflammatory regulators (*TRIM38*, *RCHY1*, *NEMF*) [22]. This convergence on adipogenesis, insulin sensitivity, and mitochondrial metabolism offers a coherent substrate for the observed metabolic adverse effect, including weight increase during clozapine treatment [24].

The target landscape of hsa-miR-4472 converges on calcium signaling and synaptic release (*HCN2*, *STIM1*, *SYT5*, *UNC13A*, *SCAMP2/5*, *SLC8A2*) and on Wnt/axon-guidance modules (*DVL3*, *RSPO4*, *PTK7*, *SEMA6A*, *EFNA3*, *PLXNA3*) [22,54,55], providing a mechanistic link to asthenia via reduced neuronal excitability and less efficient signal transmission. Additional involvement of PI3K–AKT/GPCR pathways (*PIK3R2*, *RGS6*, *GPR146/173*, *NTSR1*) and mitochondrial/proteostasis processes (*SLC25A44*, *MRPL* family, *TRIM*/*RCHY1*/*UBE2Z*) may compound fatigue through energetic stress and perturbed vesicular trafficking [24,54].

The main limitation of our study is the small sample size. TRS is heterogeneous, and a cohort this limited may not capture potential subtypes with distinct miRNA signatures. The absence of a control group (e.g., healthy volunteers or TRS patients treated with other antipsychotics) limits our ability to separate clozapine-specific effects from disease-related miRNA changes. Clozapine dosing was empirical; patients received different doses, and we did not measure plasma clozapine concentrations. We also omitted a washout period before clozapine initiation to preserve the study’s naturalistic design. Uncontrolled factors—such as diet and smoking—could have influenced miRNA profiles, although all patients were inpatients under similar conditions. Such factors may confound miRNA expression, as drug concentrations can affect epigenetic regulation. Although the inpatient setting, monotherapy design, and within-subject (delta) analyses reduce several sources of confounding, we did not protocolize quantitative measures of smoking (e.g., cotinine), detailed diet, or non-psychotropic concomitant medications.

We did not perform clozapine plasma monitoring or pharmacokinetic phenotyping. Treatment lasted four weeks; longer exposure may reveal additional miRNA changes. We analyzed only qualitative ADR characteristics and did not assess severity. At this stage, we did not evaluate the target genes regulated by the miRNAs identified in our study. Although we used standardized miRNA-seq protocols, batch effects or technical variability during library preparation may have introduced noise. Replication in independent cohorts using alternative platforms (e.g., quantitative PCR) is therefore required. While we identified candidate miRNAs, we did not experimentally validate their target genes or pathways. Future work should integrate transcriptomics with functional assays to clarify the underlying mechanisms.

## 5. Conclusions

As a pilot investigation, this study has inherent limitations, including a small sample size, the absence of a control group, and the exploratory nature of the statistical analyses. Accordingly, our findings should be considered hypothesis-generating and require further validation.

Despite these limitations, our pilot study shows an altered miRNA expression profile in TRS patients treated with clozapine. We identified possible candidate miRNAs that may serve as biomarkers of clozapine efficacy and/or safety. However, the functional roles of many of these miRNAs remain incompletely defined and warrant further study. These findings highlight the promise of pharmacoepigenetic approaches in psychiatry. Future work with larger cohorts, appropriate control groups (including healthy individuals and patients treated with alternative antipsychotics), and systematic assessment of demographic and clinical variables—together with detailed analyses of the pharmacogenetic targets of the identified miRNAs—could substantially advance personalized psychiatry.

## Figures and Tables

**Figure 1 cimb-47-01020-f001:**
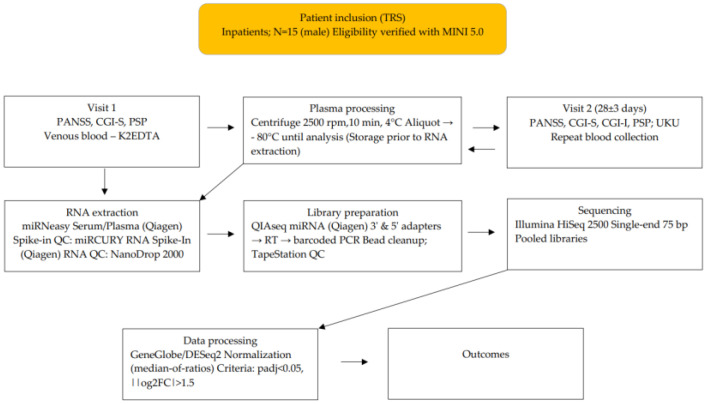
Study design and workflow.

**Figure 2 cimb-47-01020-f002:**
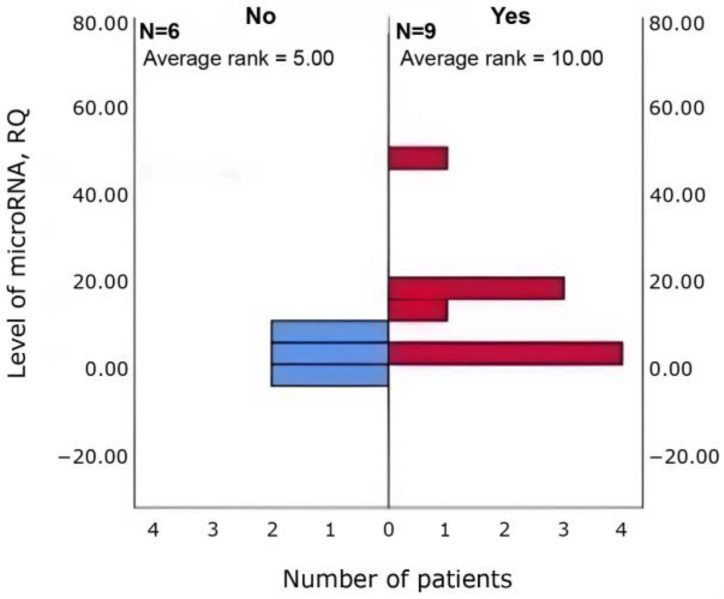
Between-group differences in Δhsa-miR-4472 in patients with vs. without asthenia, lethargy and increased fatigue (Mann–Whitney U; *p* = 0.036).

**Figure 3 cimb-47-01020-f003:**
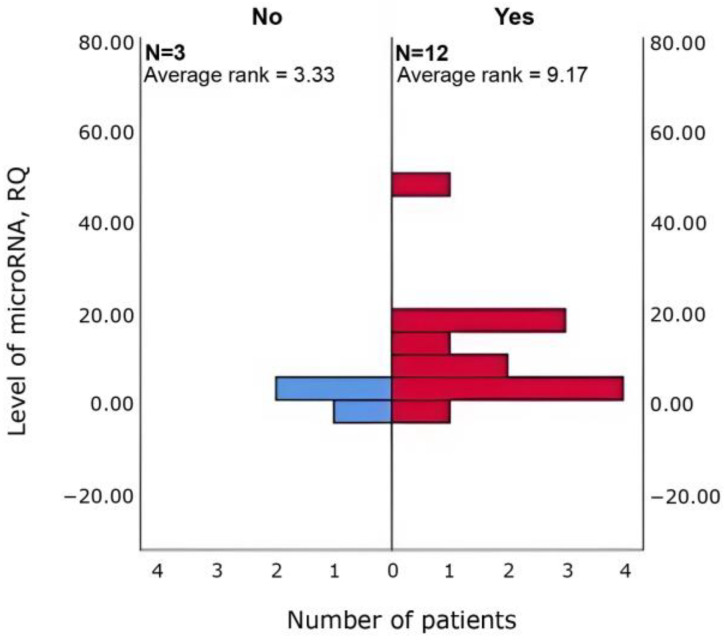
Between-group differences in Δhsa-miR-4472 in patients with vs. without increased sleep duration (Mann–Whitney U; *p* = 0.048).

**Figure 4 cimb-47-01020-f004:**
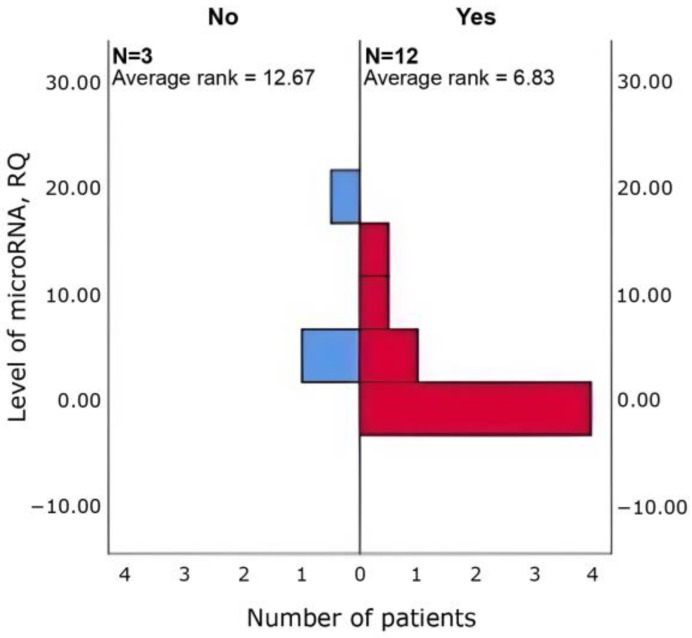
Between-group differences in Δhsa-miR-4510 in patients with vs. without increased sleep duration (Mann–Whitney U; *p* = 0.048).

**Figure 5 cimb-47-01020-f005:**
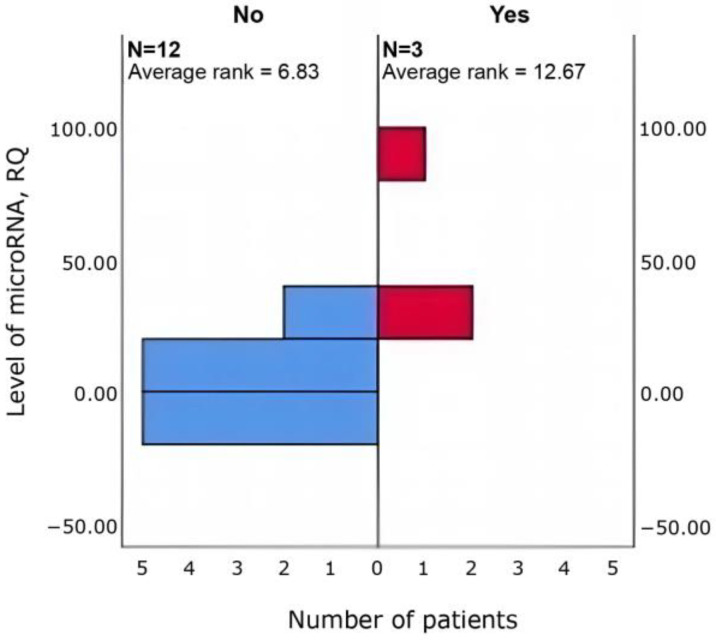
Between-group differences in Δhsa-miR-615-3p in patients with vs. without tachycardia (Mann–Whitney U; *p* = 0.048).

**Figure 6 cimb-47-01020-f006:**
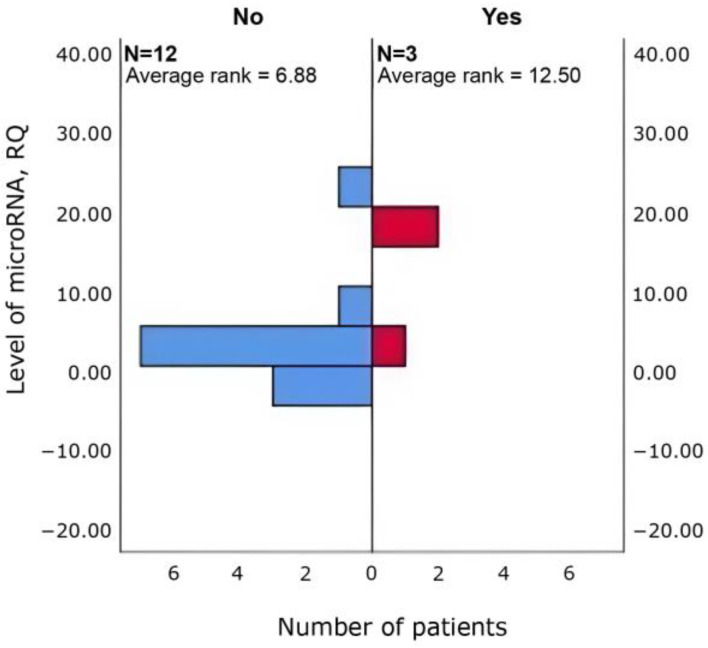
Between-group differences in Δhsa-miR-4715-3p in patients with vs. without tachycardia (Mann–Whitney U; *p* = 0.048).

**Figure 7 cimb-47-01020-f007:**
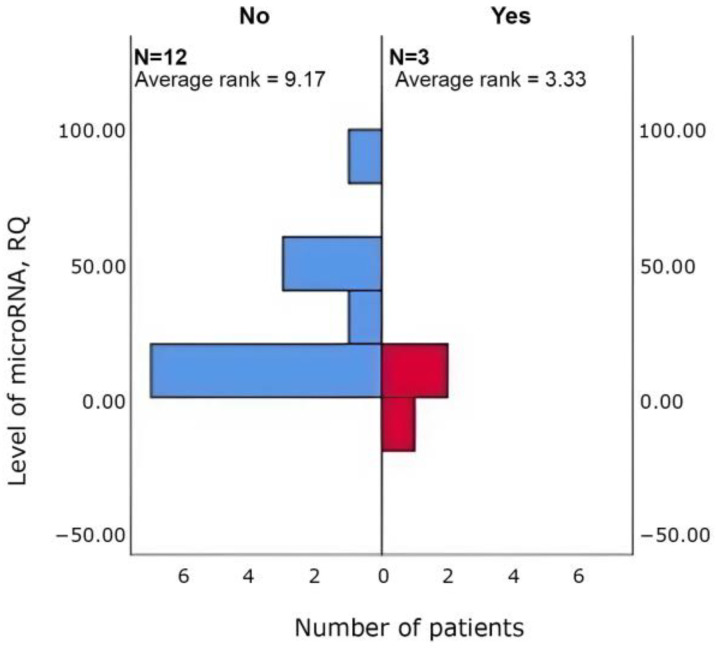
Between-group differences in Δhsa-miR-329_1_5p in patients with vs. without weight gain (Mann–Whitney U; *p* = 0.048).

**Table 1 cimb-47-01020-t001:** Differentially expressed miRNAs.

miRNA	log2FoldChange	*p* _adj_
hsa-miR-451a	−23.65	0.0001
hsa-miR-129-5p	−3.31	0.046
hsa-miR-203b-5p	1.29	0.047
hsa-miR-6873-3p	1.47	0.008
hsa-miR-9902-1	2.06	0.001
hsa-miR-9902-2	2.06	0.001
hsa-miR-615-3p	2.39	0.015
hsa-miR-4510	2.78	0.077
hsa-miR-7847-3p	2.82	0.008
hsa-miR-4472	2.92	0.003
hsa-miR-4449	3.23	0.001
hsa-miR-6509-5p	3.47	0.031
hsa-miR-6512-5p	3.58	0.033
hsa-miR-6814-5p	3.60	0.046
hsa-miR-6068	3.62	0.047
hsa-miR-7853-3p	3.85	0.026
hsa-miR-4715-3p	3.93	0.008
hsa-miR-329-1-5p	4.26	0.046
hsa-miR-6078	4.43	0.007
hsa-miR-3121-3p	4.85	0.009
hsa-miR-548i	4.90	0.020
hsa-miR-4255	5.77	0.001

**Table 2 cimb-47-01020-t002:** miRNAs associated with clozapine efficacy and safety.

Clozapine Safety	Clozapine Efficacy
Asthenia hsa-miR-4472	General efficacy hsa-miR-6814-5p
Increased sleep duration hsa-miR-4472 hsa-miR-4510	Positive symptoms hsa-miR-129-5p hsa-miR-6068 hsa-miR-6814-5p
Tachycardia hsa-miR-615-3p hsa-miR-4715-3p	General psychopathological symptoms hsa-miR-128-1-5p
Weight gain hsa-miR-329-1-5p	

**Table 3 cimb-47-01020-t003:** Functional analysis of miRNAs associated with clozapine efficacy and safety.

miRNA	Term Name	*p*-Value
Clozapine efficacy		
hsa-miR-129-5p hsa-miR-6068 hsa-miR-6814-5p hsa-miR-128-1-5p	Axon guidance	2.16 × 10^−16^
Clozapine safety		
hsa-miR-4472	Axon guidance	5.55 × 10^−8^
	ErbB signaling pathway	0.00000113
	Calcium signaling pathway	0.00000785
	Ras signaling pathway	0.000416
	Signaling pathways regulating pluripotency of stem cells	0.00777
	Wnt Signaling Pathway	0.00863
hsa-miR-4510	Axon guidance	4.07 × 10^−11^
	Signaling pathways regulating pluripotency of stem cells	4.52 × 10^−8^
	Adrenergic signaling in cardiomyocytes	0.00000108
	Synaptic vesicle cycle	0.00000214
	Wnt signaling pathway	0.00000418
	ErbB signaling pathway	0.0000438
	Calcium signaling pathway	0.0000424
	Ras signaling pathway	0.000183
hsa-miR-615-3p	Analysis yielded no significant results	
hsa-miR-4715-3p	Apelin signaling pathway	0.0000363
	Calcium signaling pathway	0.000618
	ErbB signaling pathway	0.000819
	FoxO signaling pathway	0.00120
hsa-miR-329-1-5p (hsa-miR-329-5p) ^1^	ErbB signaling pathway	1.17 × 10^−7^
	Axon guidance	5.94 × 10^−7^
	Signaling pathways regulating pluripotency of stem cells	0.00000770
	Wnt signaling pathway	0.000275
	Metabolic pathways	0.000879
	Prolactin signaling pathway	0.00312
	Cholesterol metabolism	0.00465
	Neurotrophin signaling pathway	0.00515

^1^ KEGG pathway analysis data for hsa-miR-329-1-5p were not available.

## Data Availability

The datasets generated and analyzed during this study are not publicly available due to ethical restrictions and patient confidentiality protections under Russian Federation laws on personal data protection (Federal Law No. 152-FZ). However, anonymized data supporting the findings may be made available upon reasonable request from qualified researchers, subject to approval by the Local Ethics Committee of the Russian Medical Academy of Continuous Professional Education (contact: rmapo@rmapo.ru). Requests should include a detailed research proposal and data protection plan.

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
