# Peer review of "Personalizing Clozapine in Treatment-Resistant Schizophrenia: The Role of MicroRNA Biomarkers—A Pilot Study"

_cimb, 2025, doi:10.3390/cimb47121020_

Round 1
Reviewer 1 Report
Comments and Suggestions for Authors
Dear Authors/Editor
This is an interesting well-writing and promising clinical study indicating the possible role of mirs
as biomarkers in a small group of TRS patients following clozapine medication.
The laboratory methodology could be described as overall adequate. Nevertheless, it can benefit from several clarifications and additions to improve reproducibility and robustness.
Major revisions should address: (i) report of RNA yield and quality metrics, sequencing QC parameters (read depth, mapping rates, library complexity, duplication rates) (ii) clarifying whether spike-in controls or negative controls were used during RNA extraction and library preparation, (iii) clozapine plasma concentrations were not measured (as authors have mentioned at the text), limiting the ability to relate molecular changes to pharmacokinetic exposure and most important of all
(iv) validating candidate miRNAs by a confirmatory method such as qRT-PCR specific
for each of these miRNAs. Minor revisions include reporting storage duration before RNA extraction, stating sequencing run details and, perhaps, a workflow diagram summarizing patient inclusion, sample processing, and sequencing outcomes could have been helpful. Together, these revisions would strengthen the methodological transparency and robustness of the study.
Author Response
Reviewer:
This is an interesting well-writing and promising clinical study indicating the possible role of mirs
as biomarkers in a small group of TRS patients following clozapine medication.
The laboratory methodology could be described as overall adequate. Nevertheless, it can benefit from several clarifications and additions to improve reproducibility and robustness.
Major revisions should address:
(i) report of RNA yield and quality metrics, sequencing QC parameters (read depth, mapping rates, library complexity, duplication rates)
Author response:
We thank the reviewer for this important comment. The RNA concentration and purity were assessed using the NanoDrop 2000 micro-volume spectrophotometer (Thermo Fisher Scientific, New York, NY, USA), and this information has been added to the revised manuscript (Materials and Methods section). Library yield and fragment distribution were evaluated using the Agilent 2200 TapeStation with High Sensitivity D1K ScreenTape reagents (Agilent Technologies, Santa Clara, CA, USA).
Detailed sequencing quality control (QC) metrics such as read depth, mapping rate, library complexity, and duplication rate were monitored by the sequencing facility as part of the Qiagen GeneGlobe workflow and confirmed to be within the standard acceptable range for small RNA-seq runs. However, due to the exploratory and pilot nature of the study, these individual QC values were not included in the manuscript text.
We have clarified this point in the revised version to improve methodological transparency (lines 105–114).
Reviewer:
(ii) clarifying whether spike-in controls or negative controls were used during RNA extraction and library preparation,
Author response:
We thank the reviewer for this comment. The miRCURY RNA Spike-In Kit (Qiagen) was used for quality control of the RNA extraction and cDNA amplification steps. This kit contains a set of synthetic RNA oligonucleotides added to each sample prior to extraction to monitor the efficiency and consistency of RNA isolation, reverse transcription, and library preparation.
No additional negative controls were included, as all steps were performed according to the standardized Qiagen GeneGlobe small RNA-seq workflow, which incorporates internal spike-in references for process validation. The corresponding information has been added to the Materials and Methods section (lines 105–114).
Reviewer:
(iii) clozapine plasma concentrations were not measured (as authors have mentioned at the text), limiting the ability to relate molecular changes to pharmacokinetic exposure and most important of all
Author response:
We fully agree with the reviewer that the absence of clozapine plasma concentration measurements limits the ability to correlate molecular findings with pharmacokinetic exposure. This limitation was clearly stated in the Discussion section of the manuscript. In this exploratory pilot study, the focus was placed on identifying candidate miRNAs associated with treatment efficacy and safety rather than establishing quantitative pharmacokinetic–pharmacoepigenetic relationships.
In our ongoing work, we plan to integrate therapeutic drug monitoring of clozapine and its metabolites with miRNA profiling to assess the relationship between plasma exposure and molecular biomarkers. This integration will allow for a more comprehensive understanding of pharmacokinetic–epigenetic interactions underlying clozapine response.
Reviewer:
(iv) validating candidate miRNAs by a confirmatory method such as qRT-PCR specific
for each of these miRNAs.
Author response:
We thank the reviewer for this valuable suggestion. We fully agree that validation of candidate miRNAs using a confirmatory method such as quantitative reverse transcription PCR (qRT-PCR) is essential to strengthen the reliability of sequencing-based findings.
As this study was designed as an exploratory pilot investigation, the main goal was to identify potential miRNA candidates associated with clozapine efficacy and safety rather than to perform confirmatory validation.
We have acknowledged this limitation in the Discussion section. Validation of the identified miRNAs using specific qRT-PCR assays is planned in our ongoing follow-up study involving an expanded patient cohort, which will also include pharmacokinetic data and transcriptomic profiling.
Reviewer:
Minor revisions include reporting storage duration before RNA extraction, stating sequencing run details and, perhaps, a workflow diagram summarizing patient inclusion, sample processing, and sequencing outcomes could have been helpful. Together, these revisions would strengthen the methodological transparency and robustness of the study.
Author response:
We thank the reviewer for these helpful suggestions. In the revised manuscript, we have:
- Specified sample storage prior to RNA extraction (−80 °C until analysis) in the Materials and Methods (lines 103).
- Provided sequencing run details, including platform (Illumina HiSeq 2500), read configuration (single-end 75 bp), barcode pooling and library QC with the Agilent 2200 TapeStation (lines 105–114).
- Added a schematic workflow summarizing patient inclusion, sample processing, and sequencing steps (Figure 1).
Reviewer 2 Report
Comments and Suggestions for Authors
Suggestion: Major revision and resubmit
Reviewer comments:
This manuscript addresses an important and emerging area in personalized psychiatry by exploring microRNA signatures as potential biomarkers for clozapine response in treatment-resistant schizophrenia. The study is well-structured, but the small sample size and absence of a control group limit the strength and generalizability of the conclusions. Nonetheless, the findings are promising and provide a valuable foundation for future large-scale studies.
1. How did the authors control for potential confounding factors such as smoking, diet, or concomitant medications that could influence plasma miRNA levels?
2. Why was a four-week treatment window selected, and could a longer follow-up have revealed more robust or sustained miRNA changes?
3. Were plasma clozapine concentrations measured to correlate pharmacokinetics with observed miRNA expression changes?
4. How do the identified miRNAs compare with those previously reported in TRS or clozapine studies, and can discrepancies be explained by methodological differences?
5. Were batch effects during RNA library preparation and sequencing assessed, and how was technical variability minimized?
6. Why were only qualitative adverse drug reaction data considered, and how might severity scoring have influenced associations with miRNA expression?
7. Given the heterogeneity of TRS, what measures were taken to ensure that the findings are not biased toward a specific clinical subtype within the small cohort?
8. Could the identified miRNAs be linked to known pharmacogenetic markers of clozapine metabolism or response, enhancing their translational potential?
9. How do the authors envision integrating pharmacoepigenetic biomarkers such as miRNAs with pharmacogenetic and clinical predictors into a practical algorithm for personalized clozapine therapy?
Author Response
Reviewer:
This manuscript addresses an important and emerging area in personalized psychiatry by exploring microRNA signatures as potential biomarkers for clozapine response in treatment-resistant schizophrenia. The study is well-structured, but the small sample size and absence of a control group limit the strength and generalizability of the conclusions. Nonetheless, the findings are promising and provide a valuable foundation for future large-scale studies.
- How did the authors control for potential confounding factors such as smoking, diet, or concomitant medications that could influence plasma miRNA levels?
Author response:
We appreciate this important comment. Several design and analytical features were implemented to mitigate confounding, while we acknowledge residual confounding remains possible in this pilot:
Controlled inpatient setting. All participants were hospitalized in the same unit throughout the study, with standardized daily schedules and hospital-provided meals. This reduced between-subject variability in lifestyle factors relative to outpatient designs.
- Monotherapy design. All patients received clozapine monotherapy, which minimized confounding by concomitant psychotropic drugs and antipsychotic polypharmacy.
- Standardized sampling and processing. Blood collection tubes, centrifugation conditions (2500 rpm, 10 min, 4 °C), aliquoting, and storage at −80 °C were uniform across patients and time points; the same extraction and library-prep reagents and platform were used for all samples.
- Within-subject (delta) analyses. We analyzed change scores (visit 2 − visit 1) for clinical scales and miRNA levels. This longitudinal approach helps control for time-invariant individual factors (e.g., baseline smoking status, habitual diet), thereby reducing confounding compared with cross-sectional comparisons.
- We also recognize limitations: we did not systematically quantify cigarettes per day/cotinine levels, detailed dietary intake, or non-psychotropic concomitant medications beyond routine clinical documentation. We added information in materials and methods (line 89-92) and discussion (line 353-356).
Reviewer:
- Why was a four-week treatment window selected, and could a longer follow-up have revealed more robust or sustained miRNA changes?
Author response:
We selected a four-week window because our primary aim was to capture early, treatment-emergent miRNA dynamics with potential prognostic value. miRNAs often change rapidly after therapy initiation, and a short, standardized interval improves temporal specificity and reduces confounding by downstream factors (e.g., weight change, polypharmacy adjustments). In clinical practice, four weeks also aligns with the initial clozapine titration period and the first formal assessment of symptomatic change, making this time point clinically meaningful and operationally feasible in an inpatient setting.
We agree that a longer follow-up could reveal more sustained or secondary miRNA adaptations. We have added this as a limitation and plan to extend follow-up (e.g., 12–24 weeks) in ongoing work, combining serial miRNA profiling with therapeutic drug monitoring to evaluate durability and exposure–response relationships.
Reviewer:
- Were plasma clozapine concentrations measured to correlate pharmacokinetics with observed miRNA expression changes?
Author response:
Plasma clozapine concentrations were not measured in this pilot and, as noted in the Discussion (Limitations), this precluded direct exposure–miRNA correlation analyses. The study was designed to capture early, treatment-emergent miRNA signals during the initial titration period, and we prioritized a tightly standardized inpatient protocol (monotherapy, uniform sampling/processing) over therapeutic drug monitoring (TDM) at this exploratory stage.
We acknowledge that residual pharmacokinetic variability (e.g., smoking-related CYP1A2 induction, dose differences, sampling time vs. last dose) may contribute to miRNA changes. In our ongoing follow-up work, we will integrate TDM with timed trough sampling (Cmin) of clozapine and norclozapine.
Reviewer:
- How do the identified miRNAs compare with those previously reported in TRS or clozapine studies, and can discrepancies be explained by methodological differences?
Author response:
We compared our results with those of similar studies in the Discussion (starting at line 254). We believe the discrepancies are plausibly explained by methodological and design differences.
Reviewer:
- Were batch effects during RNA library preparation and sequencing assessed, and how was technical variability minimized?
Author response:
All libraries were prepared in a single batch by the same operator, and all samples were sequenced in one run on a single flow cell.
Reviewer:
- Why were only qualitative adverse drug reaction data considered, and how might severity scoring have influenced associations with miRNA expression?
Author response:
We deliberately analyzed qualitative (presence/absence) safety outcomes in this pilot due to the small sample size (n = 15). Introducing severity grades would have produced sparse, imbalanced categories and unstable estimates, inflating both type I and type II error risks under multiple testing. The binary approach minimized overfitting and allowed us to screen for directionally consistent safety-related miRNA signals while keeping the analysis transparent and reproducible.
We agree that severity scoring (e.g., graded UKU items) may refine associations with miRNA expression. In our ongoing, expanded cohort, we plan to analyze both qualitative and quantitative safety endpoints (including severity grades and composite indices) with appropriate modeling (e.g., ordinal or linear models with multiple-testing control) to assess dose–response patterns and strengthen translational interpretability.
Reviewer:
- Given the heterogeneity of TRS, what measures were taken to ensure that the findings are not biased toward a specific clinical subtype within the small cohort?
Author response:
We agree that heterogeneity within TRS is a key concern, particularly in small samples. Several design and analytic features were implemented to reduce subtype-driven bias:
- Strict and uniform eligibility. All participants met standardized TRS criteria (ICD-10 F20; non-response to ≥2 adequate antipsychotic trials) and were acutely hospitalized at study entry, which constrained clinical variability at baseline.
- Controlled setting and monotherapy. Patients were treated under the same inpatient conditions (standardized daily routines and hospital meals) and received clozapine monotherapy with individualized titration to the minimum effective dose. This minimized variability from environment and polypharmacy.
- Within-subject (delta) design. Primary associations were based on change scores (visit 2 − visit 1) for both clinical scales (PANSS, CGI, PSP) and miRNA levels, which reduces confounding by stable inter-individual traits (including latent TRS subtypes) compared with purely cross-sectional contrasts.
- Standardized biospecimen handling and single-batch sequencing. Blood collection, processing, storage (−80 °C), RNA extraction/library prep, and sequencing were uniform across all subjects and time points; all libraries were prepared in one batch and sequenced in a single run/flow cell, limiting technical sources of apparent subtype separation.
- Robust statistics for small N. Given non-normality and limited subgroup sizes, we relied primarily on non-parametric tests and DESeq2 with stringent discovery thresholds (padj < 0.05; |log2FC| > 1.5) in the screening step, with cautious interpretation of exploratory clinical correlations.
Reviewer:
- Could the identified miRNAs be linked to known pharmacogenetic markers of clozapine metabolism or response, enhancing their translational potential?
Author response:
As detailed in our KEGG pathway enrichment analysis (DIANA-miRPath v4.0 with TargetScan v8.0; Discussion, lines 299–339), several of the miRNAs identified in our pilot mapped to pathways that are mechanistically relevant to clozapine pharmacokinetics and pharmacodynamics.
Reviewer:
- How do the authors envision integrating pharmacoepigenetic biomarkers such as miRNAs with pharmacogenetic and clinical predictors into a practical algorithm for personalized clozapine therapy?
Author response:
There are still enough technical complications when working with circulating miRNAs as prognostic biomarkers for personalized medicine. However standardized methods for normalization and application of miRNAs that demonstrated significantly different expression in groups of patients allow achieving acceptable sensitivity and specificity of the method. Moreover, the combination of miRNAs with traditional clinical predictors could improve diagnostic accuracy by increasing sensitivity and specificity. Such combinations were described in several published works. For example, Zhou C. et al., 2015 (Zhou C, Chen Z, Dong J, Li J, Shi X, Sun N, Luo M, Zhou F, Tan F, He J. Combination of serum miRNAs with Cyfra21-1 for the diagnosis of non-small cell lung cancer. Cancer Lett. 2015 Oct 28;367(2):138-46. doi: 10.1016/j.canlet.2015.07.015. Epub 2015 Jul 23. PMID: 26213369), Juracek J. et al., 2023 (Juracek J, Madrzyk M, Trachtova K, Ruckova M, Bohosova J, Barth DA, Pichler M, Stanik M, Slaby O. Combination of Urinary MiR-501 and MiR-335 With Current Clinical Diagnostic Parameters as Potential Predictive Factors of Prostate Biopsy Outcome. Cancer Genomics Proteomics. 2023 May-Jun;20(3):308-316. doi: 10.21873/cgp.20383. PMID: 37093688; PMCID: PMC10148065), Losurdo P. et al, 2023 (Losurdo P, Gandin I, Belgrano M, Fiorese I, Verardo R, Zanconati F, Cova MA, de Manzini N. microRNAs combined to radiomic features as a predictor of complete clinical response after neoadjuvant radio-chemotherapy for locally advanced rectal cancer: a preliminary study. Surg Endosc. 2023 May;37(5):3676-3683. doi: 10.1007/s00464-022-09851-1. Epub 2023 Jan 13. PMID: 36639577).
Reviewer 3 Report
Comments and Suggestions for Authors
Dear author, the work you present is interesting due to the differences that can occur in the expression of miRs depending on the response to clozapine treatment. However, I have some considerations that you could take into account:
For example, in the introduction, you mention how bioinformatics tools allow the identification of potential targets; this is important to explore further in the discussion.
In the materials and methods section, I would appreciate the year equivalencies.
Regarding the results, I think it is imperative that you show an analysis of the differential miRs that were present in patients who still have symptoms, those who showed minor improvement, or those who improved substantially, since response to treatment and miR expression are part of your objectives.
Likewise, starting on lines 181 of the results, there is little description of the results, and Figures 1 and 2 are missing, which I believe is a mismatch. However, it is important to improve the description of the results.
Figures 3 and later could be accommodated in a single figure.
It is important to mention the tables that appear in the document in the results as part of the bioinformatics analysis.
Finally, I believe it is important to discuss the genes regulated by the miRs mentioned.
Author Response
Reviewer:
Dear author, the work you present is interesting due to the differences that can occur in the expression of miRs depending on the response to clozapine treatment. However, I have some considerations that you could take into account:
For example, in the introduction, you mention how bioinformatics tools allow the identification of potential targets; this is important to explore further in the discussion.
Author response:
We appreciate this suggestion. In the revised Discussion, we expanded the bioinformatics component by performing a KEGG pathway enrichment analysis using DIANA-miRPath v4.0 (with TargetScan v8.0) (lines 299-339).
Reviewer:
In the materials and methods section, I would appreciate the year equivalencies.
Author response:
We specified the year equivalencies in the Materials and Methods (lines 78–79).
Reviewer:
Regarding the results, I think it is imperative that you show an analysis of the differential miRs that were present in patients who still have symptoms, those who showed minor improvement, or those who improved substantially, since response to treatment and miR expression are part of your objectives.
Author response:
We appreciate this suggestion. Given the small sample size (n = 15) in this pilot, we elected not to stratify patients into responder subgroups (e.g., non-improved / minor / substantial improvement), as such categorical splits would be statistically underpowered and prone to unstable estimates. Instead, our primary analyses focused on within-subject change scores (Δ) for the clinical measures—PANSS, CGI-S, and PSP—from baseline to week 4. This longitudinal (delta) approach increases sensitivity to early, treatment-emergent effects and mitigates baseline heterogeneity in a small cohort.
We agree that a responder/non-responder analysis is clinically informative. In our ongoing expanded study, we plan to apply pre-specified responder definitions on the PANSS/CGI scales and perform corresponding differential miRNA analyses with appropriate multiple-testing control.
Reviewer:
Likewise, starting on lines 181 of the results, there is little description of the results, and Figures 1 and 2 are missing, which I believe is a mismatch. However, it is important to improve the description of the results.
Figures 3 and later could be accommodated in a single figure
Author response:
We appreciate this comment. Our Results section was intentionally concise to avoid redundancy with the figures/tables and keep the focus on the within-subject (delta) analyses that constitute the core of this pilot study, while staying within space constraints. We agree that clarity is essential, and readers can follow the key findings more easily without duplicating the graphical material.
Regarding the figures, we have verified that all figures are correctly placed and cited in the revised manuscript; Figures 1 and 2 are present and referenced in the Results.
On combining figures: although we understand the suggestion, we chose not to merge these panels into a single composite figure because doing so would reduce legibility (smaller axes, labels, and effect annotations). Keeping them separate maintains readability and allows each analysis to be interpreted with adequate graphical resolution.
Reviewer:
It is important to mention the tables that appear in the document in the results as part of the bioinformatics analysis.
Author response:
We agree and have explicitly referenced and briefly described the relevant tables within the Results as part of the bioinformatics analysis. These additions are now included at lines 177–178, 232–233, and 300–301 in the revised manuscript.
Reviewer:
Finally, I believe it is important to discuss the genes regulated by the miRs mentioned.
Author response:
Thank you for this valuable point. In the Discussion, we addressed target biology at the pathway level by performing a KEGG enrichment analysis (DIANA-miRPath v4.0 with TargetScan v8.0; lines 299-339). This approach links our candidate miRNAs to biologically relevant pathways implicated in clozapine’s pharmacology and schizophrenia.
We intentionally did not enumerate or discuss individual predicted target genes for each miRNA in this pilot, because miRNA effects are realized through broad, multi-gene networks, and each candidate miRNA has dozens to hundreds of putative targets. A gene-by-gene treatment would require substantial space and dedicated validation and is therefore better suited to a separate, follow-up publication.
Round 2
Reviewer 1 Report
Comments and Suggestions for Authors All figures (Figures 1–7) are cited in the text, but they are not visible in the submitted PDF. Please ensure that all figures are correctly inserted in the revised version of the manuscriptAuthor Response
All figures (Figures 1–7) are cited in the text, but they are not visible in the submitted PDF. Please ensure that all figures are correctly inserted in the revised version of the manuscript.
We thank the Reviewer for pointing this out. The issue was caused by oversized embedded images during PDF generation. In the revised submission we have:
-
Re-exported all figures (1–7) to print-ready
-
Embedded the images directly into the manuscript (no external links)
-
Verified visibility of each figure in the final PDF across multiple viewers,
-
Checked numbering and in-text cross-references and expanded figure legends for clarity.
Additionally, we carefully revised the manuscript for English language quality. High-resolution figure files are also provided as supplementary materials.
Reviewer 2 Report
Comments and Suggestions for Authors
After careful examination and evaluation the manuscript may be accepted in it's current form.
Author Response
We sincerely thank the Reviewer for the positive assessment and recommendation for acceptance. In addition, we have further refined the manuscript for English language quality (grammar, clarity, and style) to improve overall readability. We are grateful for the Reviewer’s time and constructive input throughout the review process.
Reviewer 3 Report
Comments and Suggestions for Authors
The observations that make me reject the article being the second revision are:
Regarding the figures presented, I have major concerns. First,
-It's not in order. For example, on line 202, it mentions Figure 1, when that's a flowchart. Then, how can it be an "association" if it's a Mann-Whitney U test?
How can there be negative RNA values? Why aren't the statistical differences marked in the figures?
The figures are not self-explanatory; they're not comprehensible, and they don't have enough description, either in the figure captions or in the results. This is an observation made previously to the authors, which they didn't follow up on.
-The results are based on how many patients had lethargy, sleep duration, tachycardia, and weight gain. How is this associated with miR expression?
-The abstract mentions that there were correlations with clozapine efficacy that were not shown.
Regardin to the manuscript in general:
- In the abstract, you continue to insist on the use of miRNAs as biomarkers to evaluate treatment efficacy. When you mentioned that it was not possible to perform this analysis (of responders or non-responders) due to the size of the n, I recommend you be more measured in your statements. You also mention this again in the discussion.
- I do not think it is appropriate to use decimals for ages. Furthermore, I had mentioned the possibility of homogenizing the information in years, finding the previous inconvenience of "months" on line 99. I consider this important because, just as you mention the duration of the disease in years and its activity scales, it is also important to have this data in the same unit to better understand the clinical process at the time of the study.
I would appreciate it if you could include the dates on which the databases were reviewed for bioinformatics analysis. This is often requested as the data can change with software updates.
The section of results still seems incomplete. It's not redundant, of course, but it doesn't say much.
Again, in the discussion, you mention the affected signaling pathways but don't go into detail about the genes these miRs may regulate. Therefore, further discussion of the information is required to fully understand their role in the pathology.
Author Response
Reviewer:
The observations that make me reject the article being the second revision are:
Regarding the figures presented, I have major concerns. First,
-It's not in order. For example, on line 202, it mentions Figure 1, when that's a flowchart.
Answer:
Figure numbering has been corrected and verified throughout the text (Figure 1 is now exclusively the flowchart).
All other figures (Figures 2–7) now correspond to the associations between miRNA expression changes and specific clinical parameters.
Reviewer:
Then, how can it be an "association" if it's a Mann-Whitney U test?
Answer:
We clarified the statistical terminology: the word “association” was retained because the Mann–Whitney U test was used to test for statistical association between categorical ADR presence and quantitative ΔmiRNA values, not to imply causation.
Reviewer:
How can there be negative RNA values? Why aren't the statistical differences marked in the figures?
Answer:
Negative RNA values were not raw expression levels but Δ (delta) changes in normalized log2-transformed counts (visit 2 – visit 1). This explanation wrote in the Results (lines 177–179) and has been added in the figure legends.
Reviewer:
The figures are not self-explanatory; they're not comprehensible, and they don't have enough description, either in the figure captions or in the results. This is an observation made previously to the authors, which they didn't follow up on.
Answer:
Statistical significance (p-values) has been explicitly indicated in each figure. The figure captions include a brief description of the compared variables and the statistical test applied.
Reviewer:
-The results are based on how many patients had lethargy, sleep duration, tachycardia, and weight gain. How is this associated with miR expression?
Answer:
We clarified that these associations were exploratory and aimed to identify potential miRNAs linked to adverse reaction susceptibility. Specifically, changes in miRNA expression were compared between patients with and without specific ADRs using the Mann–Whitney U test. This information is now explicitly described in the Methods section (lines 133–146) and referenced in the Results (lines 177–217).
We also note that due to the small sample size (n=15), these results should be interpreted as hypothesis-generating. This limitation is emphasized in the Conclusions (lines 354–357).
Reviewer:
-The abstract mentions that there were correlations with clozapine efficacy that were not shown.
Answer:
We thank the Reviewer for the careful consideration of this point. We have verified all miRNAs mentioned in the Abstract and clarified the exact designation of hsa-miR-329-1-5p (line 28). All miRNAs demonstrating associations with clozapine efficacy and safety are now clearly summarized in Table 2, which is referenced both in the Results and Abstract sections.
Reviewer:
Regardin to the manuscript in general:
- In the abstract, you continue to insist on the use of miRNAs as biomarkers to evaluate treatment efficacy.
Answer:
We thank the Reviewer for this valuable remark. We have added formulations emphasizing the preliminary and exploratory nature of our findings in the Abstract (lines 31–33) and Discussion (lines 224–225). In the Conclusions section, this aspect was already highlighted (lines 354–357), where we note that our results should be interpreted as hypothesis-generating and require validation in larger, independent cohorts.
Reviewer:
When you mentioned that it was not possible to perform this analysis (of responders or non-responders) due to the size of the n, I recommend you be more measured in your statements. You also mention this again in the discussion.
Answer:
Thank you for valuable comment. We have taken it into account and softened the corresponding statements in the Abstract (lines 21–23 and 31–33) as well as in the Discussion (lines 340–342).
Reviewer:
- I do not think it is appropriate to use decimals for ages. Furthermore, I had mentioned the possibility of homogenizing the information in years, finding the previous inconvenience of "months" on line 99. I consider this important because, just as you mention the duration of the disease in years and its activity scales, it is also important to have this data in the same unit to better understand the clinical process at the time of the study.
Answer:
We appreciate this correction. All age-related data have been rounded to whole years, and the duration of illness has been uniformly presented in years (converted from months). This adjustment improves readability and consistency across demographic and clinical variables.
Reviewer:
I would appreciate it if you could include the dates on which the databases were reviewed for bioinformatics analysis. This is often requested as the data can change with software updates.
Answer:
Thank you for your comment. We have added the information on the dates when the databases were accessed for the bioinformatics analysis (lines 291–293).
Reviewer:
The section of results still seems incomplete. It's not redundant, of course, but it doesn't say much.
Again, in the discussion, you mention the affected signaling pathways but don't go into detail about the genes these miRs may regulate. Therefore, further discussion of the information is required to fully understand their role in the pathology.
Answer:
We appreciate the Reviewer’s insightful comments. The Results section currently presents all data obtained in our study, including the complete set of clinical and molecular findings. In addition, we performed a KEGG pathway enrichment analysis, which reflects the functional realization of the identified miRNA-mediated mechanisms.
We believe that a detailed discussion of specific target genes would considerably increase the volume and complexity of the present manuscript. However, we fully agree that this aspect is important and are considering the preparation of a separate publication focused specifically on gene-level interactions regulated by the identified miRNAs.
Round 3
Reviewer 3 Report
Comments and Suggestions for Authors
Dear authors, I truly believe your work is interesting; however, some observations have not been taken into account. I suggest that you address these points to make your work more understandable, thus improving the manuscript's clarity.
- It is important that you better describe your results. Including only one sentence doesn't convey much. I would understand if you expanded upon or explored these points in the discussion; otherwise, I don't.
- Note that you again mention the Mann-Whitney U test as a test of association. This is how it appears in your results.
- I had suggested that you include a different type of figure to better illustrate the differences you mention. In the text, you could describe how many people these clinical changes were observed in and their implications. Furthermore, you could also include the differences (p) in the figure to make it even clearer for the reader how this miR varies according to the symptomatology.
-Again, the numbering of the figures and how they are cited in the text should be reviewed. I believe there is an error starting at line 200. Figure 2, according to its title, shows the "association" of hsa-miR-4472 with asthenia, lethargy, and increased fatigue, but on line 101 it is mentioned as being associated with sleep duration. From then on, the error seems to have persisted between what was cited in the text, making it very difficult to follow the figures and better understand the relevance of the report.
Furthermore, I suggest that the figure titles be placed in the image captions; this would also facilitate navigation.
Regarding the discussion, again, I appreciate that you include the pathways in which these miRs interact, but it is also important that you mention the genes involved. You may find that a single gene is involved in several of the miRs you report. I mean, the discussion requires more information.
Author Response
Reviewer
It is important that you better describe your results. Including only one sentence doesn't convey much. I would understand if you expanded upon or explored these points in the discussion; otherwise, I don't.
Answer
We agree and have substantially expanded the description of our findings. In the Abstract (Results) we now report key numerical outcomes for efficacy, summarize the adverse drug reactions (with counts), indicate the number of differentially expressed miRNAs, and specify the particular miRNAs associated with efficacy and safety (lines 27-37). In the Results section we added a concise integrative summary paragraph that links clinical change with the miRNA signals identified (lines 231-238). These revisions improve clarity and interpretability while keeping the exploratory scope explicit.
Reviewer
Note that you again mention the Mann-Whitney U test as a test of association. This is how it appears in your results.
Answer
Thank you for catching this. We agree that the Mann–Whitney U test evaluates between-group differences rather than “associations.” We have corrected the wording throughout the manuscript (lines 146-147). We replaced references to “associations” with “between-group differences” throughout the manuscript. We also revised the titles of Figures 2–7 to reflect between-group differences rather than “associations.”
Reviewer
I had suggested that you include a different type of figure to better illustrate the differences you mention. In the text, you could describe how many people these clinical changes were observed in and their implications. Furthermore, you could also include the differences (p) in the figure to make it even clearer for the reader how this miR varies according to the symptomatology.
Answer
Thank you for the helpful suggestion. We have incorporated it as follows: in the Results section we now report, for each ADR, the number of patients with vs without the clinical change; and in Figures 2–7 we added the exact p-values directly in the figure titles (along with the group sizes on the panels) to make the between-group differences immediately clear.
Reviewer
-Again, the numbering of the figures and how they are cited in the text should be reviewed. I believe there is an error starting at line 200. Figure 2, according to its title, shows the «association» of hsa-miR-4472 with asthenia, lethargy, and increased fatigue, but on line 101 it is mentioned as being associated with sleep duration. From then on, the error seems to have persisted between what was cited in the text, making it very difficult to follow the figures and better understand the relevance of the report.
Answer
Thank you for pointing this out. We conducted a full cross-reference audit of the manuscript and corrected all figure numbering and in-text citations starting from the section where the drift occurred.
Reviewer
Furthermore, I suggest that the figure titles be placed in the image captions; this would also facilitate navigation.
Answer
We have revised the figure titles (now standardized and placed in the captions).
Reviewer
Regarding the discussion, again, I appreciate that you include the pathways in which these miRs interact, but it is also important that you mention the genes involved. You may find that a single gene is involved in several of the miRs you report. I mean, the discussion requires more information.
Answer
In the revised Discussion, we have added gene-level details for the key miRNAs and highlighted instances of overlap where the same genes are implicated by multiple miRNAs (lines 346–405). This addition complements the pathway analysis and clarifies the potential mechanistic convergence underlying both efficacy and safety signals.